# Arbuscular Mycorrhizal Symbiosis Differentially Affects the Nutritional Status of Two Durum Wheat Genotypes under Drought Conditions

**DOI:** 10.3390/plants11060804

**Published:** 2022-03-17

**Authors:** Valentina Fiorilli, Moez Maghrebi, Mara Novero, Cristina Votta, Teresa Mazzarella, Beatrice Buffoni, Stefania Astolfi, Gianpiero Vigani

**Affiliations:** 1Department of Life Sciences and Systems Biology, Università degli Studi di Torino, 10124 Torino, Italy; valentina.fiorilli@unito.it (V.F.); moez.maghrebi@unito.it (M.M.); mara.novero@unito.it (M.N.); cristina.votta@unito.it (C.V.); teresa.mazzarella@unito.it (T.M.); beatrice.buffoni@unito.it (B.B.); 2Department of Agricultural and Forestry Sciences (DAFNE), University of Tuscia, 01100 Viterbo, Italy; sastolfi@unitus.it

**Keywords:** durum wheat, drought, nutrient homeostasis, arbuscular mycorrhizal fungi symbiosis

## Abstract

Durum wheat is one of the most important agricultural crops, currently providing 18% of the daily intake of calories and 20% of daily protein intake for humans. However, being wheat that is cultivated in arid and semiarid areas, its productivity is threatened by drought stress, which is being exacerbated by climate change. Therefore, the identification of drought tolerant wheat genotypes is critical for increasing grain yield and also improving the capability of crops to uptake and assimilate nutrients, which are seriously affected by drought. This work aimed to determine the effect of arbuscular mycorrhizal fungi (AMF) on plant growth under normal and limited water availability in two durum wheat genotypes (Svevo and Etrusco). Furthermore, we investigated how the plant nutritional status responds to drought stress. We found that the response of Svevo and Etrusco to drought stress was differentially affected by AMF. Interestingly, we revealed that AMF positively affected sulfur homeostasis under drought conditions, mainly in the Svevo cultivar. The results provide a valuable indication that the identification of drought tolerant plants cannot ignore their nutrient use efficiency or the impact of other biotic soil components (i.e., AMF).

## 1. Introduction

Due to climate change, crop productivity across the world is threatened and there is an urgent need for new solutions for crop adaptation to environmental changes [1,2,3]. Drought stress is considered one of the major environmental factors concerning world food security [4]. It affects enzyme activity, nutrient uptake, assimilation, and crop production [5,6,7,8]. The intensity and frequency of drought periods are being increased by climate change [9,10,11], implicating new challenges for agricultural practice and the need to better understand the mechanisms of plant adaptation and tolerance in order to ensure food production can continue into the future [11].

Durum wheat (*Triticum durum* Desf.) is one of the world’s most consumed cereal grains and is widely cultivated in the Mediterranean region and other semiarid regions across the world [12,13], where changes in precipitation patterns have resulted in drought and critical agricultural loss [14,15]. Plants can respond to drought stress by developing morphological, physiological, and biochemical changes and various adaptive mechanisms to mitigate against dehydration [16,17,18], thereby allowing them to increase drought tolerance.

Additionally, drought may cause nutrient deficiencies, even in fertilized fields [19]. The mechanisms that plants have evolved for nutrient uptake, translocation, and assimilation may not function optimally under drought conditions. It is generally accepted that fertilization is most effective when plants are not water-stressed and that irrigation is most effective when nutrients are not limited.

Considering sustainable wheat production, the use of drought tolerant wheat genotypes seems to be the most appropriate choice, but the development of proper agronomic approaches could also be critical. Therefore, there is a need to look for adequate approaches to enhance the drought resistance of plants to improve crop yield [20]. A possible approach for integrated management systems is soil inoculation with plant growth promoting microbes, such as arbuscular mycorrhizal fungi (AMF), which help to alleviate the detrimental effects of drought stress [21,22]. In general, the symbiotic interaction of plants with AMF has been defined as an essential functional system to improve the nutrient uptake and growth of host plants [23,24]. In addition, many studies have elucidated the physiological and molecular mechanisms exerted by AMF on nutrient and water uptake in plants [25]. For this reason, AM symbiosis is currently considered to play an instrumental role within sustainable agriculture in terms of moderating water deficit stress by improving plant–water relations and the mineral nutritional status and thus, enhancing crop productivity under drought conditions [26,27,28,29]. Several studies performed on different host plants, such as wheat, barley, maize, strawberry, and onion, have shown clear evidence of drought stress mitigation using AMF [30,31,32,33]. The improved adaptation of mycorrhized plants could be initially related to the AMF extraradical hyphal network that is connected by plant roots, which ensures the extensive exploration of soil volume and supplies the plant with water and mineral nutrients [34,35]. Previous studies have reported the different mechanisms that take place in mycorrhizal plants to overcome drought stress, including the enhancement of water use efficiency (WUE) [36] and the increase in antioxidant activity [37]. In addition, AM symbiosis regulates the physiological performance of plants through the alteration of their hormonal balance [38], osmotic adjustment [39], and the improvement of their nutrient uptake and assimilation [40].

The impact of mycorrhizal symbiosis on plant mineral nutrition has been well investigated and, in particular, it has been demonstrated that plants possess a symbiotic phosphate uptake pathway [41,42]. Other studies have characterized ammonium transporters that are exclusively expressed in cells containing arbuscules: the key structure of symbiosis when nutrient exchange occurs between partners [43,44,45]. AMF are also capable of significantly improving plant sulfur (S) acquisition. Allen and Shachar-Hill [46] have shown that the AMF *Rhizophagus irregularis* takes up sulfate and amino acids containing sulfur and transfers them into the plant. On the plant side, studies of the transcriptomic response of *Medicago truncatula* have shown that the regulation of sulfate transporters depends on S concentration and mycorrhizal colonization [47]. Similarly, other reports have demonstrated the impact of AM symbiosis on the transfer of S from the fungus to the plant [48] and have identified a sulfate transporter expressed in arbusculated cells that are specifically involved in S uptake [49].

It is well known that S nutrition may affect dough and baking quality through its effects on the composition of wheat grain proteins [50]. Indeed, most of the sulfate taken up by plants is used for the synthesis of amino acids containing sulfur, cysteine (Cys) and methionine (Met) [51], which are the major components of proteins [52] and represent about 20% of the grain and play a key role in dough functionality. In particular, Cys helps to improve the elasticity of gluten because it is responsible for the disulfide bond in proteins, which affects the processing quality of the wheat [53,54]. Furthermore, there has been evidence of the important role of sulfur and compounds containing sulfur within abiotic stress defenses, including defenses against drought stress [55,56]. In contrast to *Triticum aestivum*, the outcomes of AM symbiosis in durum wheat have been poorly investigated so far. This study aimed to fill this gap in knowledge by investigating the impact of AM symbiosis on the nutritional status of durum wheat under drought stress. Considering that AM symbiosis responsiveness is subject to plant genetic diversity [57], two durum wheat cultivars with different genetic backgrounds were investigated. We selected a commercial durum wheat variety (Svevo) and an ancient variety (Etrusco, *Triticum turanicum turgidum*), which has been cultivated in Italy since the time of the Etruscans, hence the name. In this work, an integrative approach was employed, combining the measurement of traits related to plant growth, the analysis of plant nutritional status, and the expression of some genes that were encoded for the key enzymes in sulfur homeostasis that are involved in the water stress response and the mycorrhizal colonization of wheat plants challenged by drought.

## 2. Results

### 2.1. AM Symbiosis Differentially Affects Plant Growth Related Traits under Drought Conditions in Svevo and Etrusco Cultivars

The two-way ANOVA within the genotype revealed that the water status significantly affected all measured traits in both cultivars (*p* < 0.001) with exception of root length (RL). Notably, AM symbiosis only significantly affected the root fresh weight (RFW) (*p* < 0.01) in Etrusco but all traits in Svevo. However, only shoot fresh weight (SFW) (*p* < 0.01) and RWC (*p* < 0.001) displayed a significant interaction between the variables (water status x AMF) in Svevo (Appendix A). The results showed that drought stress (CS) significantly affected the fresh weights of roots (RFW) and shoots (SFW), shoot length (SL), and the relative water content (RWC) of both cultivars compared to the watered plants (CW) (Figure 1A–E). On the other hand, under drought stress, inoculation with AMF (MS) did not alter the plant growth traits of either cultivar compared to the control plants (CS). Interestingly, under stressed conditions, RWC increased in mycorrhized Svevo plants, whereas it did not change in the leaves of mycorrhized Etrusco plants (MS) compared to those of the control plants (CS) (Figure 1E). A pair comparison analysis (*t*-test) between the two cultivars revealed that under the control well-watered (CW) conditions, RL was significantly higher in Svevo than in Etrusco, whereas the SL was significantly lower. Under the control stressed (CS) conditions, RFW and SFW were significantly lower in Svevo than in Etrusco. Under the mycorrhized well-watered (MW) conditions, SL, SFW, and RWC were significantly lower in Svevo than in Etrusco. Under the mycorrhized stressed (MS) conditions, RL, RFW, and SFW were significantly lower in Svevo than in Etrusco; by contrast, the RWC of Svevo was significantly higher (Figure 1A–E).

### 2.2. Expression of Genes Involved in Water Stress Response in Mycorrhizal and Control Plants

To analyze the molecular pattern of the drought responsiveness of the cultivars in the presence or absence of AM fungal colonization, the transcripts of the central genes that are involved in water stress response (*SHN1* and *DRF1*) were investigated in the roots of plants grown under the different conditions. The two-way ANOVA carried out within the genotype revealed that the water status significantly affected the relative transcript level of *TdSHN1* and *TdDRF1* in both cultivars. In addition, AMF significantly affected the relative expression of *TdSHN1* in Etrusco. Moreover, the interaction between variables (water status X AMF) significantly affected the expression of *TdDRF1* in both cultivars and *TdSHN1* in Svevo (Appendix A). Under the control stressed conditions (CS), the steady state level of the transcripts of *TdSHN1* was significantly upregulated in both varieties, although more evidently in Svevo (Figure 2A). However, a different behavior was observed for *TdDRF1*, whose relative transcript was not significantly affected by drought stress (CS) in either cultivar (Figure 2B). Additionally, under drought stress, the relative transcript level of *TdSHN1* was not significantly affected by inoculation with AMF (MS) in either cultivar compared to the control plants (CS). On the contrary, the relative transcript abundance of *TdDRF1* in Svevo plants was significantly increased under mycorrhized stressed conditions (MS). A pair comparison analysis (*t*-test) between the two cultivars revealed that the relative expression of *TdSHN1* was significantly higher in Svevo than in Etrusco under the control stressed (CS) conditions (Figure 2A). Interestingly, both analyzed genes had a significantly higher expression in Svevo than in Etrusco under mycorrhized stressed (MS) conditions (Figure 2A,B).

### 2.3. Evaluation of Mycorrhizal Colonization Level in Both Cultivars and Water Regimes

To investigate the differences between the mycorrhizal colonization levels of the cultivars and determine whether water stress could influence AM symbiosis functionality, the expression of *TdPT11* (the putative homolog of the phosphate transporter (PT) gene considered as an AM functional marker gene) and the fungal housekeeping gene *Fm18S* were evaluated (Figure 3). The two-way ANOVA revealed that water status and genotype affected the *Fm18S* and *TdPT11* variables. Moreover, *Fm18S* and *TdPT11* displayed a significant interaction between variables (water status X genotype) (Appendix A). The steady state level of the *Fm18S* transcripts was significantly increased in both cultivars under mycorrhized stressed conditions (MS) (Figure 3A), while *TdPT11* was not affected (Figure 3B). A pair comparison analysis (*t*-test) revealed that the relative expression of *Fm18S* was significantly higher in Svevo than Etrusco under mycorrhized watered (MW) conditions (Figure 3A). In addition, under the mycorrhized stressed (MS) conditions, the relative expression of both analyzed genes was higher in Svevo (Figure 3A,B).

The expression of *TdPT11* reveals the arbuscular functionality and can be roughly related to the A% value (Figure 4B), which describes the arbuscular abundance within the root system, while the expression of *Fm18S* can be roughly related to F% (mycorrhization frequency) (Figure 4A).

In both water regimes, the expression of *TdPT11* was in line with the A% value (Figure 4B). In MW, the arbuscular abundance was comparable in both cultivars, while in MS, the slight increase in arbuscular abundance that was detected by the morphological assay was supported by *TdPT11* upregulation in the Svevo cultivar. The expression of *Fm18S* and the value of F% were in agreement in MS conditions, indicating that Svevo was more highly colonized than Etrusco under water stress. By contrast, the morphological and molecular values in MW conditions were different. This discrepancy could be due to the fact that the *Fm18S* expression level represented both fungal extraradical and intraradical mycelium since the extraradical mycelium was not removed during the root sampling, while the morphological assay exclusively quantified the fungal intraradical structures (i.e., intraradical hyphae, arbuscules, vesicles)

### 2.4. Drought Differentially Affected AMF Colonization in Svevo and Etrusco

The two-way ANOVA revealed that water status significantly affected all of the mycorrhization indicators, while genotype only significantly affected mycorrhization frequency (F%) (*p* < 0.01). However, only mycorrhization frequency (F%) (*p* < 0.001) displayed a significant interaction between variables (water status x genotype) (Appendix A). The results showed that under mycorrhized stressed conditions (MS), mycorrhization frequency (F%) decreased in Etrusco plants while it was not affected in Svevo plants (Figure 4A). On the other hand, the arbuscular abundance (A%) decreased in both cultivars under mycorrhized stressed conditions (MS) (Figure 4B). A pair comparison analysis (*t*-test) between the two cultivars revealed that under MW conditions, F%, which evaluates the diffusion of AM fungal hyphae within the entire root system, was significantly lower in Svevo than in Etrusco. On the contrary, under MS conditions, Svevo displayed a significantly higher diffusion of AM fungal hyphae (F%) (Figure 4A).

### 2.5. Drought Differentially Affected Leaf Nutritional Status in Svevo and Etrusco under Mycorrhizal and Control Conditions

To investigate the impact of water regimes and AM symbiosis on the nutritional status of durum wheat cultivars, cation (potassium (K^+^), sodium (Na^+^), calcium (Ca^2+^), and magnesium (Mg^2+^)) and anion (chloride (Cl^−^), nitrate (NO_3_^−^), sulfate (SO_4_^2−^), and phosphate (PO_4_^3−^)) quantification was performed in leaf tissues.

The PCA performed on the cation content revealed that the first and second components (PC1 and PC2) accounted for about 82% of the total variability (PC1: 57% and PC2: 25%) (Figure 5A). The separation observed between the SMS plants and the SCS plants along PC2 was related to the variation in Na^+^ content. Additionally, a separation of the SCS plants from the SCW plants was evident along PC2, suggesting that water status impacted K^+^, Ca^2+^, and Mg^2+^ concentrations (Figure 5A, Appendix A). The two-way ANOVA carried out within the genotype revealed that water status significantly affected Na^+^ and Ca^2+^ content in both cultivars and K^+^ (*p* < 0.001) and Mg^2+^ (*p* < 0.01) concentrations in Svevo. Notably, AM symbiosis only significantly affected K^+^ (*p* < 0.001) and Na^+^ (*p* < 0.001) content in Svevo. However, water status x AMF interaction significantly affected Na^+^ (*p* < 0.001), Ca^2+^ (*p* < 0.05), and Mg^2+^ (*p* < 0.05) concentrations in Svevo (Appendix A). Furthermore, the two-way ANOVA conducted under water conditions showed that genotype only significantly affected K^+^ (*p* < 0.01) content in well-watered conditions and AMF only significantly affected Na^+^ (*p* < 0.001) content in drought conditions. Interestingly, Etrusco did not display any variation in the concentration of cations between the different conditions. In Svevo plants, the content of K^+^, Ca^2+^, and Mg^2+^ decreased under drought stress conditions (CS). On the other hand, under drought stress, inoculation with AMF (MS) significantly increased the Na^+^ content in Svevo plants compared to control plants (CS) (Figure 5C). A pair comparison analysis (*t*-test) between the two cultivars revealed that K^+^ content was significantly higher in Svevo than in Etrusco under CW and CS conditions (Figure 5B). Notably, Na^+^ concentration was significantly higher in Etrusco than in Svevo under the CS conditions. Contrarily, Na^+^ concentration was higher in mycorrhized stressed (MS) Svevo plants. The two-way ANOVA within the genotype revealed that water status significantly affected the Na^+^/K^+^ ratio in both cultivars, while AMF only significantly affected the Na^+^/K^+^ ratio (*p* < 0.001) in Svevo. Moreover, a significant interaction between the variables (water status × AMF) was observed (*p* < 0.001) in Svevo (Appendix A).

The PCA performed on the concentrations of the inorganic anions identified two principal components, PC1 and PC2, which accounted for 53% and 25% of the variance, respectively (Figure 6A). The PCA scatterplot identified the clear separation of the mycorrhized stressed (SMS) Svevo plants from the other groups along PC1. This separation was mainly attributable to the NO_3_^−^ and PO_4_^3−^ concentration variation among the samples (PC1) (Figure 6A, Appendix A). Additionally, the separation of the SMS plants from the SMW plants along PC2 suggested that the water status impacted SO_4_^2−^ concentration. The two-way ANOVA carried out within the genotype revealed that water status significantly affected SO_4_^2−^ concentration in both Etrusco (*p* < 0.01) and Svevo (*p* < 0.001) plants and NO_3_^−^ (*p* < 0.001) and PO_4_^3−^ (*p* < 0.05) concentrations in Svevo only. Notably, AM symbiosis significantly affected all anion contents in Svevo and only SO_4_^2−^ (*p* < 0.001) and PO_4_^3−^ (*p* < 0.05) concentrations in Etrusco. However, SO_4_^2−^ concentration displayed a significant interaction between the variables (water status × AMF) in both cultivars, with Cl^−^ (*p* < 0.001) content in Etrusco and NO_3_^−^ (*p* < 0.001) concentration in Svevo (Appendix A). The results showed that Cl^−^ concentration only decreased under drought stress (CS) in Etrusco (Figure 6B). Furthermore, SO_4_^2−^ concentration for both cultivars was higher in the leaves of mycorrhized watered plants (MW) than in the other studied conditions (Figure 6C). The concentrations of NO_3_^−^ and PO_4_^3−^ were not affected by the conditions for Etrusco plants, while AMF improved the NO_3_^−^ and PO_4_^3−^ concentrations in Svevo under drought stressed conditions compared to control stressed conditions (CS) (Figure 6D,E). A pair comparison analysis (*t*-test) between the two cultivars revealed that the SO_4_^2−^ content in Svevo was significantly higher than in Etrusco in all experimented conditions (Figure 6C). Under the control stressed (CS) conditions, Cl^−^ concentration in Svevo was significantly higher than in Etrusco (Figure 6B). In addition, NO_3_^−^ and PO_4_^3−^ concentrations were significantly higher in Svevo than in Etrusco under mycorrhized stressed (MS) conditions (Figure 6D,E) (*p* < 0.05). To monitor the sulfur nutritional status, Sorin et al. [58] and Etienne et al. [59] proposed and validated the ([Cl^−^] + [NO_3_^−^] + [PO_4_^3−^]): [SO_4_^2−^] ratio as a relevant indicator of the S nutritional status (hereafter defined as the S index) in different plant species. The two-way ANOVA within the genotype revealed that the water status variable did not affect the S index in either cultivar, while AMF symbiosis significantly impaired it. Moreover, a significant interaction between the variables (water status X AMF) was only observed in Etrusco (Appendix A). In mycorrhized watered (MW) conditions, both cultivars displayed the lowest S index values, indicating that the presence of AMF under well-watered conditions provides a beneficial effect on the sulfur nutritional status of plants. However, the Etrusco cultivar displayed significantly higher S index values in mycorrhized stressed (MS) conditions than Svevo (Figure 6F).

### 2.6. Expression of Genes Involved in Sulfur Homeostasis

Due to the difference observed between the levels of sulfate concentration in the leaves of the two cultivars, the effect of drought stress and AMF on sulfate uptake and assimilation pathways were investigated by analyzing the expression of gene coding for major transporters (high-affinity sulfate transporters *TdSULTR1.1* and *TdSULTR1.3*) and the enzymes involved in sulfur uptake, translocation, and assimilation (O-acetylserine (thiol) lyase, *TdOASTL1*, and serine acetyltransferase, *TdSAT1*, respectively). The two-way ANOVA within the genotype revealed that water status significantly affected the relative transcript levels of *TdSAT1* and *TdSULTR1.3* in both cultivars and *TdSULTR1.1* in Etrusco only. AMF significantly affected the relative expression of *TdSAT1* and *TdSULTR1.3* in both cultivars and *TdOASTL1* in Etrusco only. Moreover, a significant interaction between the variables (water status × AMF) was observed for the expression of (i) *TdSAT1* in both varieties and (ii) *TdSULTR1.1* and *TdSULTR1.3* in Etrusco (Appendix A). The relative transcript abundance of *TdSULTR1.1* in Etrusco was upregulated under drought conditions (CS), while it was not affected in Svevo (Figure 7A). Interestingly, in well-watered conditions, mycorrhizal plants (MW) displayed a transcript level increase in *TdSultr1.3* for both cultivars compared to the control conditions (CW) (Figure 7B). Additionally, under drought stress, the relative expression of *TdSULTR1.3* and *TdSAT1* increased in MS Svevo cultivars (Figure 7B) and in both MS cultivars (Figure 7D), respectively, compared to the control plants (CS). However, the relative expression of the *TdOASTL1* gene was unaffected in both cultivars (Figure 7C). A pair comparison analysis (*t*-test) between the two cultivars revealed that the relative expressions of *TdSULTR1.1* and *TdSULTR1.3* were significantly higher in Svevo than in Etrusco under control well-watered (CW) conditions (*p* < 0.05) (Figure 7A,B). Interestingly, all of the genes considered were significantly higher in Svevo than in Etrusco under mycorrhized stressed (MS) conditions, except for the *TdSULTR1.1* gene (Figure 7).

## 3. Discussion

### 3.1. Drought-Induced Responses of Durum Wheat Genotypes Are Differentially Modulated by AMF

Growth traits, such as length and biomass, have been typically used as selection criteria for wheat drought tolerance [60]. In the present study, both wheat cultivars were affected by drought stress and displayed a significant reduction in the fresh weights of roots (RFW) and shoots (SFW), shoot length (SL), and relative water content (RWC) compared to the watered plants. These results were consistent with previous studies reporting a drought-induced decrease in most of the growth parameters of durum wheat [61,62,63]. Growth inhibition under water deprivation could be mainly attributed to photosynthesis limitation resulting from stomata closure and commonly leads to dry biomass and height reduction [60,64]. Remarkably, inoculation with AMF did not produce any alterations in the plant growth traits of either cultivar under drought stress compared to the control plants (CS) according to Recchia et al. [65]. Similarly, Tarnabi et al. [66] revealed that bread wheat inoculated with AMF induced a decrease in the shoot and root length during water stress.

Relative water content (RWC) is considered an indicator of the water stress tolerance of plants under drought conditions [67]. In the present study, the RWC was significantly decreased by drought stress in both cultivars. However, under this condition, Svevo displayed a higher RWC compared to Etrusco, indicating a higher water retention capability in the former with respect to the latter cultivar. Several studies on durum wheat have shown that the RWC decreases in water deficit conditions [68,69]. Svevo and Etrusco cultivars grown under drought stress showed differential behaviors depending on their differential water status. Interestingly, under stressed conditions, the AMF inoculation of Svevo plants significantly promoted water retention, as revealed by the significantly higher RWC in the leaves of mycorrhized Svevo plants compared to those of the control Svevo plants, suggesting that the higher level of AM colonization detected in MS roots might limit drought stress severity in the Svevo cultivar. These findings agreed with previous reports showing that mycorrhization improved the water uptake of plants under drought conditions [70,71,72]. Additionally, different reports have suggested that the higher RWC in mycorrhized plants is either related to the ability of soil growing hyphae to increase the absorption region of the host plant roots and the absorption of water with low potential from the rhizosphere [17,72,73] or to the higher ability of the plant to control water loss through stomatal regulations [36] to maintain RWC under severe drought conditions. In particular, Zarik et al. [67] suggested different potential mechanisms that could be responsible for the RWC increase mediated by mycorrhizal inoculation, such as (i) stomatal regulation, (ii) higher stomatal conductance and transpiration fluxes, (iii) the indirect effect of improved nutrient uptake, (iv) more significant osmotic adjustment, and/or (v) higher root hydraulic conductivity than control plants. Our results indicated that mycorrhized Svevo plants displayed a higher drought tolerance than Etrusco, which was probably due to the higher susceptibility of the Svevo genotype to perform AM symbiosis under drought stress. Notwithstanding the higher RWC, mycorrhized stressed (MS) Svevo plants showed a decreased root and shoot biomass compared to those grown under the control stressed conditions, suggesting that the combined effect of AM symbiosis and water stress negatively impact Svevo fitness, contrary to the results obtained previously by other works [71,74]. However, Etrusco did not reveal such a decreased growth, suggesting that the phenotype observed in Svevo might be attributed to its genetic background.

The interaction of plant roots with AMF may induce an alteration in the expression of the genes involved in biotic and abiotic resistance responses [39]. During droughts, plant hormones, such as abscisic acid (ABA), are crucial for regulating stress tolerance by inducing the closing of stomata, thus minimizing transpiration water loss [75]. Furthermore, ABA is also necessary to alleviate stress damage by activating many stress-responsive genes that encode enzymes for the biosynthesis of osmotically active metabolites and LEA proteins [75,76]. However, to investigate the molecular patterns of the drought responsiveness, especially under AMF colonization, we analyzed the expression levels of SHN1 (WAX INDUCER1/SHINE1 (WIN1/SHN1)) and DRF1 (dehydration responsive factor 1), two of the genes involved in water stress perception [77]. Under control stressed conditions, *TdSHN1* was upregulated in both varieties, although more evidently in Svevo. However, mycorrhized stressed Svevo plants displayed the highest induction compared to the control stressed plants, highlighting the beneficial AMF effect on *TdSHN1* expression. Djemal and Khoudi [78] provided evidence that *TdSHN1* is involved in multiple abiotic stress tolerance processes and the expression of *TdSHN1* is strongly induced by drought in durum wheat plants. In Arabidopsis, the *SHN1* gene was found to improve water use efficiency by modifying leaf diffusive properties due to the accumulation of high levels of wax [79]. Therefore, our results suggested that the beneficial role of AMF in Svevo plants might be related to the improvement of water use efficiency through the modulation of *SHN1* genes.

Another related gene belonging to the dehydration responsive element binding protein (DREB) family that has been reported and characterized in durum wheat is *TdDRF1*, and its expression is related to water deficit response [80]. In the present study, *TdDRF1* was also upregulated in the presence of AMF in Svevo plants under drought stress. Previous reports have observed that mycorrhization increased the ABA concentration in plants and that this increase could modulate AMF symbiosis by affecting the formation of arbuscules and helping the colonization of plant roots [76,81], especially under stress conditions in which ABA plays a role in plant protection and defense [31]. Despite the absence of direct evidence, the differential behaviors of the mycorrhized roots of Svevo and Etrusco plants might be related to the different ABA contents compared to the control plants, which could, in turn, limit water loss in mycorrhized plants and induce the expression of drought responsive genes. Accordingly, Recchia et al. [65] found similar results by identifying interesting common bean genes that were exclusively regulated in response to water deficit during the AMF inoculation of bean roots.

### 3.2. AMF Affects the Nutritional Status of Durum Wheat Genotypes under Drought Conditions

Drought stress significantly hinders nutrient uptake and assimilation, thereby affecting plant growth [7,8]. In general, AMF colonization can improve plant nutritional status by increasing nutrient acquisition in plants [82,83] and also in nutrient deficient soils [83,84]. Accordingly, different studies have confirmed the contribution of AMF colonization in overcoming drought stress by improving plant nutrition, which could be considered as an important drought tolerance mechanism [85,86]. Plant tolerance to drought could be primarily due to the capability of the fungal extraradical mycelium to increase the access of the plants to soil water [34,87,88,89].

Several previous studies have suggested that inoculation with AMF considerably improves the uptake of nitrogen and phosphorus in host plants [42,90,91,92,93]. In this study, mycorrhized Svevo plants accumulated considerably higher mineral nutrient leaf concentrations (SO_4_^2−^, NO_3_^−^, PO_4_^3−^, and Na^+^) than the control Svevo plants under drought stress. The enhanced acquisition of these nutrients may be due to the role of considerable AMF hyphae in mycorrhized plants improving the absorption surface area compared to the roots alone [72]. Our results agreed with the previous findings that mycorrhized plants displayed a significantly higher N and P than the control plants under drought stress and an enhanced drought stress tolerance [94,95]. For mycorrhized plants under drought stress, different reports have suggested that the increased stomatal conductance, leaf water potentials, chlorophyll content, and photosynthetic efficiency are due to improved nutritional status [96,97,98]. On the other hand, K^+^ plays an essential role in the stress tolerance of plants by being involved in stomatal movement and the amelioration of deleterious effects [7,8]. However, in the present study, the tolerance of the mycorrhized plants against drought stress was unlikely to be related to K^+^ accumulation, even in the presence of AMF. Moreover, the Mg^+^ content in the leaves was not affected by AMF symbiosis under drought conditions.

Interestingly, sulfate concentration was higher in the leaves of mycorrhized watered plants (both Etrusco and Svevo) than in other conditions. Allen and Shachar-Hill [46] reported a symbiotic uptake pathway for sulfate. Despite under drought conditions, such AMF-mediated sulfate increase was not observed in both cultivars, the sulfate concentration still displayed higher value in Svevo than in Etrusco. Furthermore, phosphate displayed the highest value concentration in in mycorrhized Svevo plants. Accordingly, previous studies in *Medicago truncatula* have demonstrated that AM symbiosis improves the sulfur nutrition in low sulfate environments [47,48] and also the plant’s phosphate status [48]. Giovannetti et al. [49] studied the S starvation responses in mycorrhized plants, which showed higher sulfate concentrations in shoots and roots, suggesting a positive role for mycorrhizal fungi in plant S nutrition at low sulfate availability. Under the mycorrhized stressed conditions, the higher concentrations of chloride found in Etrusco plants might be related to the low sulfate levels, as suggested by Coubert et al. [99]. To maintain osmotic balance, plants must modulate the content of inorganic anions, which act as osmolytes in the cells. Therefore, in S starving conditions, plants tend to balance anion content by modulating chloride, phosphate, and nitrate contents through a mechanism known as homeostasis co-limitation [99]. The S index variation highlights how this inorganic anion balance occurs in a specific tissue, mirroring the possible nutrient mobilization process from the tissue itself to other organs, such as spikes. In particular, mycorrhized plants displayed a decreasing S index (meaning a better sulfur nutrient status of the plants) under well-watered and water stressed conditions, indicating that AMF positively impacts sulfur homeostasis in Svevo and Etrusco. Our results agreed with the beneficial role of AM symbiosis in the nutrient homeostasis of plants [40,67,100,101]. Notably, the Etrusco cultivar displayed higher S index values under all conditions considered, suggesting that sulfur homeostasis is differentially affected in Etrusco with respect to the Svevo cultivar.

Sulfur and compounds containing sulfur are considered important elements in plant tolerance to abiotic stress [55] and in wheat, they also play an essential role in dough functionality, thus affecting bread and pasta quality [54]. As observed in our results, AMF can improve sulfate accumulation under drought conditions, but limited information about the molecular basis of S homeostasis in mycorrhized plants is available. For that, we assessed the expression of some sulfur related genes that are involved in sulfate uptake, translocation, and assimilation. Concerning sulfate uptake and translocation, we investigated the two high-affinity sulfate transporters *TdSULTR1.1* and *TdSULTR1.3* in durum wheat cultivars. *SULTR1.1* is considered the most responsive gene coding for the high-affinity transporter to the sulfate availability in durum wheat, while *SULTR1.3* is a high-affinity sulfate transporter involved in loading sulfate into phloem and distributing it to sink organs [102,103].

Interestingly, when grown in well-watered conditions, mycorrhizal plants achieve higher sulfate concentrations in the leaves of both cultivars than in control conditions, reflecting the induced transcript levels of the sulfate transporter *SULTR1.3*. Under drought, the AMF-dependent upregulation of *SULTR1.3* was higher in Svevo than in Etrusco cultivars. Previous reports have revealed that plant roots colonized with AM fungi induced the upregulation of SULTR genes at low sulfate concentrations to enable them to survive in such environments [47,48]. Additionally, the relative transcript abundance of *SULTR1.1* in Etrusco was strongly upregulated under drought conditions, while it was not affected in Svevo. As previously suggested, in mycorrhized plants, organic S compounds can be transferred *via* the mycorrhizal uptake pathway to other transporter systems [48]. The upregulation of *SULTR1.1* in Etrusco could be driven by a higher drought-induced sulfate demand. Previous reports have demonstrated that the high-affinity *SULTR1.1* transporter is induced under S deficiency to ensure sulfate uptake continues [104,105]. Similarly, maize plants grown under drought conditions displayed the induction of *SULTR1.1* expression levels in roots [56,106]. In addition, based on our S index data, we suggest that sulfur homeostasis was differentially affected in Etrusco compared to Svevo and that the same occurred for the expression of *SULTR1.1* and *SULTR1.3* genes. These findings were consistent with the idea that the capability of a given cultivar to adjust its S homeostasis could rely on the induction of sulfur uptake from the media or on the SO_4_^2−^ mobilization within the plant. The observed changes in S metabolic demand reasonably suggest an increased S assimilation rate. The synthesis of cysteine is considered as an essential step in the assimilatory sulfur pathway by the enzymes serine acetyltransferase (SERAT) and O-acetylserine (thiol) lyase (OAS-TL) via the intermediate O-acetylserine (OAS) [56]. During drought stress, the colonization of the roots of Svevo by AMF induced the expression of *TdSAT1*. This upregulation could increase the level of OAS, which positively affected sulfate assimilation and could enhance sulfate uptake and reduction to synthesize cysteine under drought stress. Kawashima et al. [107] reported SAT-induced expression in sulfur-starved conditions. In contrast, the expression of the *TdOASTL1* gene was unaffected in both cultivars. These results agreed with previous studies showing that the expression of *TdOASTL1* and OASTL activity were unaffected in durum wheat plants under sulfur deprivation [98]. Similarly, Krueger et al. [108] demonstrated that OASTL activity was not significantly affected by sulfur deprivation in plants.

## 4. Materials and Methods

### 4.1. Plant Material and Growth Conditions

The seeds of two durum wheat (*Triticum durum* Desf.) cultivars (Svevo and Etrusco) were allowed to germinate in Petri dishes for 7 days, in the dark and on wet filter paper. The seedlings were then transferred to pots containing (i) a mixture (7:3) of quartz sand and a fungal inoculum composed of *Funneliformis mosseae* (formerly *Glomus mosseae*, BEG 12) spores, extraradical mycelium, and colonized Sorghum roots along with the cultivation substrate of Sorghum plant (MycAgro Lab, Bretenière, France) for mycorrhizal conditions (M), (ii) a mixture (7:3) of quartz sand and a carrier composed of Sorghum roots along with the cultivation substrate of the Sorghum plant, instead of the fungal inoculum, for non-mycorrhizal (C) conditions.

The pots were then watered with tap water until the fully wet condition was reached (100%) and each pot was weighed to record the substrate water capacity (SWC%), as described in Lehnert et al. [109]. The plants were grown in a controlled growth chamber (16 h light at 23 °C and 8 h dark at 21 °C) and watered when needed with a modified Long Ashton (LA) solution (3.2 μM Na_2_HPO_4_·12 H_2_O [110]). A low P content was used to ensure good levels of root colonization since a higher concentration could impair the establishment of AM symbiosis. Every week, all pots were weighed to monitor the SWC% and the water stress (S conditions) was gradually induced by lowering the SWC%. Untreated plants (W conditions) were watered with the Long Ashton solution, as described above. The weight of the pot was maintained at >90% in well-watered plants and was reduced to 70% in stressed plants.

The experimental setup consisted of four conditions for both cultivars: (i) control (non-mycorrhizal) well-watered (CW); (ii) control (non-mycorrhizal) stressed (CS); (iii) mycorrhized watered (MW); (iv) mycorrhized stressed (MS).

For each condition, five independent biological replicates were prepared. All of the plants were sampled after 56 days of cultivation. At harvest, the roots and shoots were separated and their relative fresh weight was measured. The root system from each mycorrhizal plant was divided into two samples: one to assess the level of mycorrhizal colonization (performed on at least 60 cm of root per sample) and the other for RNA extraction. To assess the level of mycorrhizal colonization, the roots were stained in 0.1% (*W/V*) cotton blue lactic acid. The level of mycorrhizal colonization was assessed according to the method of Trouvelot et al. [111] and four parameters were obtained (F%, mycorrhization frequency; M%, AMF colonization intensity; A%, arbuscular abundance; a%, arbuscules in the colonized portion of the root system). The roots were immediately frozen in liquid nitrogen for the molecular analyses and stored at −80 °C.

### 4.2. RNA Extraction and qRT-PCR Analysis

The total RNA was extracted from the roots of mycorrhizal and non-mycorrhizal plants using the Plant RNeasy Kit (Qiagen, Hilden, Germany), according to the manufacturer’s instructions. According to the manufacturer’s instructions, samples were also treated with TURBO DNase (Ambion, Austin, TX, USA). The extracted RNA was quantified using Nanodrop with the incorporated ND 1000 software. The RNA samples were routinely checked for DNA contamination through PCR analysis using the housekeeping gene GAPDH (Appendix A). For single-strand cDNA synthesis, 11 μL of the total RNA were denatured at 65 °C for 5 min and then reverse transcribed at 25 °C for 10 min, 42 °C for 50 min, and 70 °C for 15 min. The final volume contained 1 μL of Randoms Primers (Sigma-Aldrich, Darmstadt, Germany) (100 ng/μL), 1 μL of dNTPs (10 mM), 4 μL of 5X First-Strand buffer, 2 μL of DTT (0.1 M), 1 μL of SuperScriptII reverse transcriptase (Invitrogen), and 11 μL of the RNA.

Quantitative RT-PCR was performed using Rotor-Gene Q (Qiagen) and the primers indicated in the Appendix A. Each PCR reaction was carried out in a total volume of 15 μL, containing 2 μL of diluted cDNA (1:10), 7.5 μL of 2X SYBER Green PCR Master Mix, and 5.5 μL of forward and reverse primer mix. We used the following PCR program: 95 °C for 90 s, 40 cycles of 95 °C for 15 s, and 60 °C for 30 s for GAPDH, TdSNH1, TdDRF1, Fm18S, and TdPT11; 58 °C for TdOASTL1 and TdSAT1; 56 °C for TdSULTR1.3; and 54 °C for TdSULTR1.1. A melting curve (80 steps with a heating rate of 0.5 °C per 10 s and a continuous fluorescence measurement) was recorded at the end of each run to exclude the generation of non-specific PCR products. We performed all of the reactions on at least five biological and two technical replicates. The baseline range and Ct values were automatically calculated using Rotor-Gene Q 5plex software. The transcript levels were normalized to the Ct value of *GAPDH*. Only Ct values leading to a Ct mean with a standard deviation below 0.5 were considered.

### 4.3. Measurement of Relative Water Content (RWC)

The relative water content (RWC) of fully developed flag leaves was determined according to Sade et al. [112]. The RWC percentage was calculated using the following formula:RWC (%) = [(FW − DW)/(SW − DW)] × 100
where FW is the leaf fresh weight, DW is the dry weight, and SW is the saturated weight. The SW (weight measured at turgidity) was determined after soaking the leaf in a CaCl_2_ (5 mM) solution for 24 h at room temperature.

### 4.4. Cation and Anion Analysis by Capillary Electrophoresis (CE)

The determination of cation and anion concentrations was carried out on the second and third leaves as representatives of the physiologically active leaves of the plants at the time of harvesting (the flag leaf was harvested for RWC determination). They were collected and washed with Milli-Q water, soft soaked with paper, immediately frozen in liquid nitrogen, and then freeze-dried. The cations and anions were then extracted with ultrapure water using a mortar and pestle. The extracts were filtered using 0.22 µm filters and assayed by capillary electrophoresis (Agilent 7100, Agilent Technologies, Santa Clara, CA, USA). Cations, such potassium (K^+^), sodium (Na^+^), calcium (Ca^2+^), and Magnesium (Mg^2+^), were analyzed using a bare fused silica capillary with an extended light path BF3 (i.d. = 50 μm, I = 56 cm, L = 64.5 cm). Sample injection was at 50 mbar for 5 s with +30 kV voltage and a detection wavelength of 310/20 nm. Anions, such as chloride (Cl^−^), sulfate (SO_4_^2−^), nitrate (NO_3_^−^) and phosphate (PO_4_^3−^), were analyzed through a bare fused silica capillary column with an extended light path BF3 (i.d. = 50 µm, I = 72 cm, L = 80.5 cm). Sample injection was followed by 50 mbar pressure for 4 s with −30 kV voltage and detection at the 350/380 nm wavelength. All cations and anions were identified using pure standards. The final cation and anion contents in each sample were calculated as µg g^−1^ DW (dry weight). The experiments were repeated at least three times.

### 4.5. Statistical Analysis

Due to the multidimensional nature of the dataset, the trait variations between the two cultivars were analyzed by principal component analysis (PCA) to identify whether certain morphological and physiological traits were associated with AM symbiosis under drought stress or not. The experimental data were subjected to two-way analyses of variance (ANOVA). The two-way ANOVAs were performed within genotypes to evaluate the significance of water status, the AMF treatment, and the interaction between water status × AMF treatment. The Tukey’s test was selected as the ANOVA post hoc test (*p* < 0.05). Student’s *t*-test was used to assess the significance of the observed differences between the Etrusco and Svevo plants (*p* < 0.05) within the different conditions. The principal component analysis and ANOVA were conducted using Past3 software.

## 5. Conclusions

Investigating how water stress affects the nutrient uptake and assimilation capability allows for the identification of crop resilience to drought and nutrient deficiency. In the present study, we investigated the different aspects of drought tolerance in mycorrhized and non-mycorrhized plants of two durum wheat cultivars that were subjected to drought stress. We observed a positive effect of mycorrhizal symbiosis on the water status and nutrient uptake of Svevo cultivars grown in drought stress conditions compared to Etrusco cultivars. Furthermore, the observed mycorrhizal responsiveness of the wheat under drought stress confirms the important role of AMF in these environments. Considering that most vegetable and seed crops are cultivated in semiarid areas and regions suffering from temporary droughts, it is important not only to ascertain how water stress affects the nutrient uptake and assimilation capability of these crops but also to identify genes and genotypes that can increase crop resilience to drought and nutrient deficiency under indisputable climate change. Our work provides evidence that AMF can have a positive or neutral impact on durum wheat under drought conditions, depending on the genotype.

## Figures and Tables

**Figure 1 plants-11-00804-f001:**
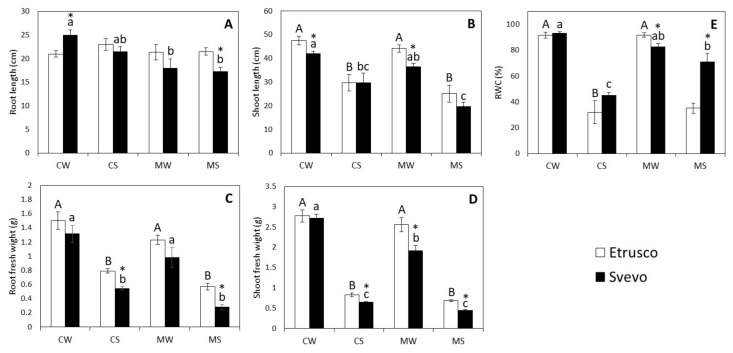
The morpho-physiological traits of Svevo and Etrusco cultivars under different conditions: Root (**A**) and shoot (**B**) length, root (**C**) and shoot (**D**) fresh weight, and relative water content (RWC) in the leaves (**E**) of two durum wheat cultivars grown under control well-watered (CW), control stressed (CS), mycorrhized watered (MW), and mycorrhized stressed (MS) conditions. Etrusco is represented by white bars and Svevo by black bars. Data are presented as the mean ± SE (*n* = 5) and significant differences between the Etrusco and Svevo plants (within conditions) are indicated as follows: * *p* < 0.05. Different letters indicate significant differences between conditions within a genotype (uppercase letters for Etrusco and lowercase letters for Svevo) (Tukey’s test, *p* < 0.05). An absence of letters means that there were no significant differences between conditions. The two-way ANOVA results are reported in Appendix A.

**Figure 2 plants-11-00804-f002:**
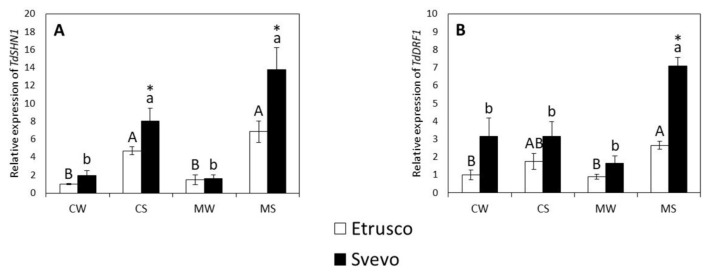
The relative expression of genes involved in the water stress response, *TdSHN1* (**A**) and *TdDRF1* (**B**), in the roots of two durum wheat cultivars grown under control well-watered (CW), control stressed (CS), mycorrhized watered (MW), and mycorrhized stressed (MS) conditions. Etrusco is represented by white bars and Svevo by black bars. Data are presented as the mean ± SE (*n* = 5) and significant differences between the Etrusco and Svevo plants (within conditions) are indicated as follows: * *p* < 0.05. Different letters indicate significant differences between conditions within a genotype (uppercase letters for Etrusco and lowercase letters for Svevo) (Tukey’s test, *p* < 0.05). The two-way ANOVA results are reported in Appendix A.

**Figure 3 plants-11-00804-f003:**
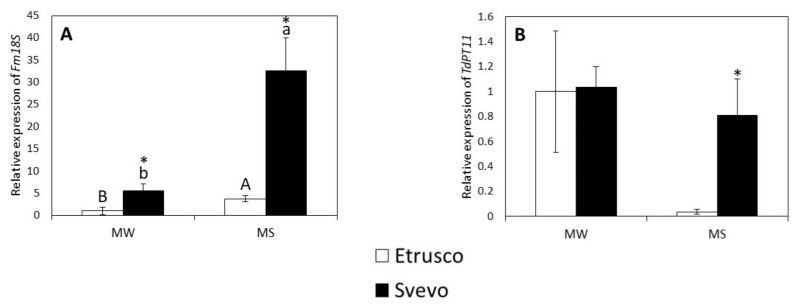
The relative expression of genes involved in mycorrhizal colonization, *Fm18S* (**A**) and *TdPT11* (**B**), in the colonized roots of two durum wheat cultivars grown under mycorrhized watered (MW) and mycorrhized stressed (MS) conditions. Etrusco is represented by white bars and Svevo by black bars. Data are presented as the mean ± SE (*n* = 5) and significant differences between the Etrusco and Svevo plants (within conditions) are indicated as follows: * *p* < 0.05. Different letters indicate significant differences between conditions within a genotype (uppercase letters for Etrusco and lowercase letters for Svevo) (Tukey’ s test *p* < 0.05). An absence of letters means that there were no significant differences between conditions. The two-way ANOVA results are reported in Appendix A.

**Figure 4 plants-11-00804-f004:**
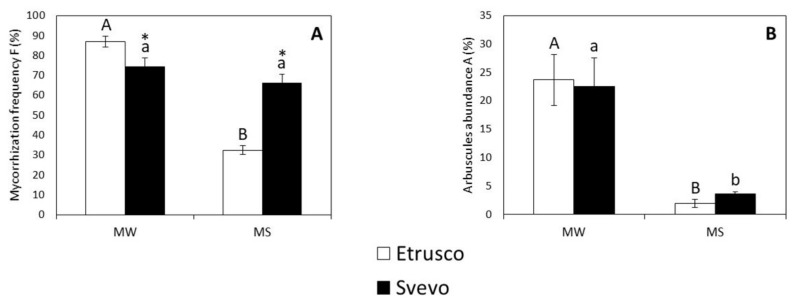
Colonization level parameters: mycorrhization frequency F% (**A**) and arbuscular abundance in the entire root system A% (**B**) within the colonized roots of two durum wheat cultivars grown under mycorrhized watered (MW) and mycorrhized stressed (MS) conditions. Etrusco is represented by white bars and Svevo by black bars. Data are presented as the mean ± SE (*n* = 5) and significant differences between the Etrusco and Svevo plants (within conditions) are indicated as follows: * *p* < 0.05. Different letters indicate significant differences between conditions within a genotype (uppercase letters for Etrusco and lowercase letters for Svevo) (Tukey’s test, *p* < 0.05). The two-way ANOVA results are reported in Appendix A.

**Figure 5 plants-11-00804-f005:**
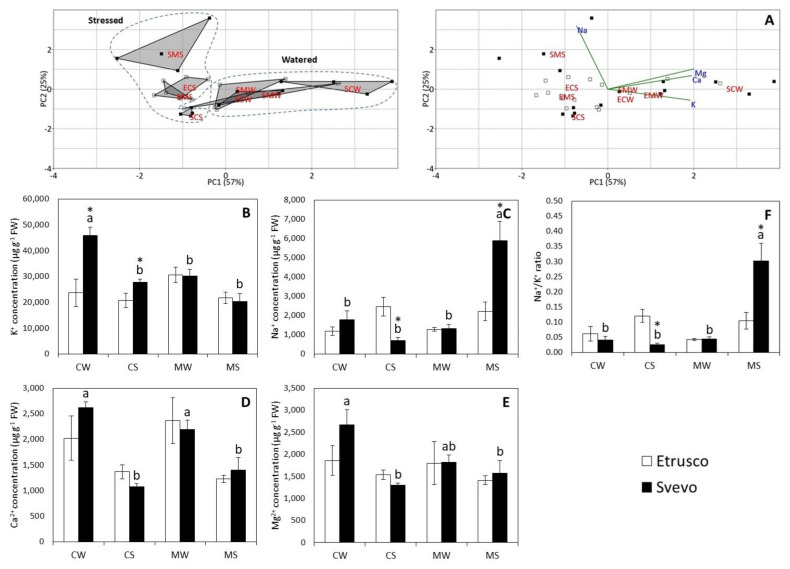
Cation concentration in Svevo and Etrusco leaves: (**A**) the principal component analysis (PCA) with scatterplot (left) and biplot (right) projections of the data onto the subspace spanned by both components (PC 1 and PC2), which are colored according to cultivars with each group labeled with its condition class. Etrusco is represented by white squares and Svevo by black squares. The potassium (K^+^) (**B**), sodium (Na^+^) (**C**), calcium (Ca^2+^) (**D**), magnesium (Mg^2+^) (**E**), and Na^+^/K^+^ ratio (**F**) in the leaves of two durum wheat cultivars grown under control well-watered (CW), control stressed (CS), mycorrhized watered (MW), and mycorrhized stressed (MS) conditions. Etrusco is represented by white bars and Svevo by black bars. Data are presented as the mean ± SE (*n* = 4) and significant differences between the Etrusco and Svevo plants (within conditions) are indicated as follows: * *p* < 0.05. Different letters indicate significant differences between conditions within a genotype (uppercase letters for Etrusco and lowercase letters for Svevo) (Tukey’s test, *p* < 0.05). An absence of letters means that there were no significant differences between treatments. The two-way ANOVA results are reported in Appendix A.

**Figure 6 plants-11-00804-f006:**
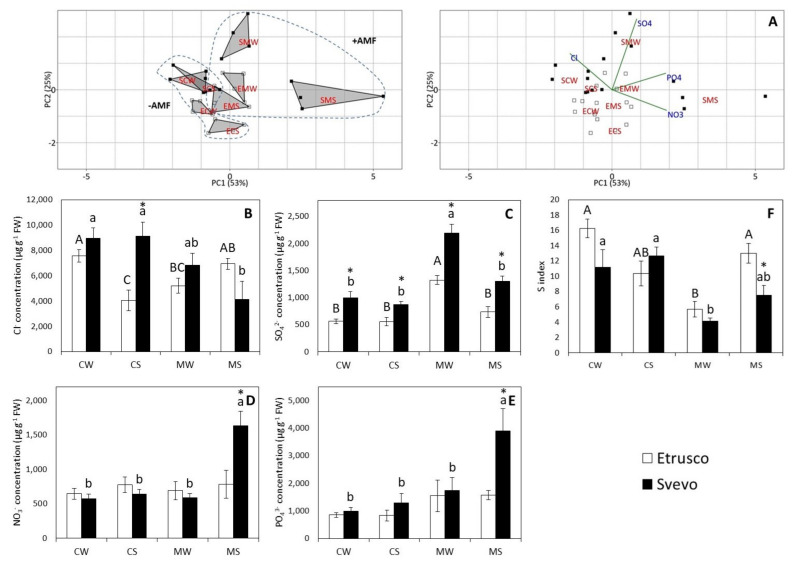
Anion concentration in Svevo and Etrusco leaves: (**A**) the principal component analysis (PCA) with scatterplot (left) and biplot (right) projections of the data onto the subspace spanned by both components (PC 1 and PC2), which are colored according to cultivars with each group labeled with its condition class. Etrusco is represented by white squares and Svevo by black squares. Abbreviations: -AMF, not mycorrhizal plants; +AMF, mycorrhizal plants. The chloride (Cl^−^) (**B**), sulfate (SO_4_^2−^) (**C**), nitrate (NO_3_^−^) (**D**), phosphate (PO_4_^3−^) (**E**), and S index (**F**) in the leaves of two durum wheat cultivars grown under control well-watered (CW), control stressed (CS), mycorrhized watered (MW), and mycorrhized stressed (MS) conditions. Etrusco is represented by white bars and Svevo by black bars. Data are presented as the mean ± SE (*n* = 4) and significant differences between the Etrusco and Svevo plants (within conditions) are indicated as follows: * *p* < 0.05. Different letters indicate significant differences between conditions within a genotype (uppercase letters for Etrusco and lowercase letters for Svevo) (Tukey’ test, *p* < 0.05). An absence of letters means that there were no significant differences between conditions. The two-way ANOVA results are reported in Appendix A.

**Figure 7 plants-11-00804-f007:**
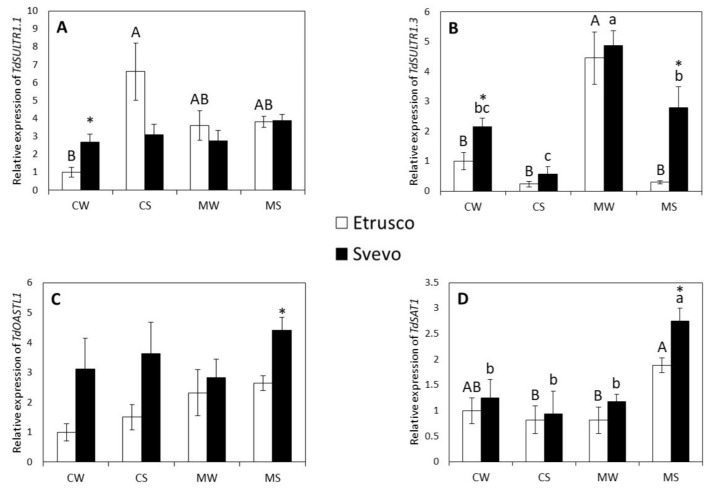
The relative expression of genes involved in sulfur metabolism, *TdSULTR1.1* (**A**), *TdSULTR1.3* (**B**), *TdOASTL1* (**C**), and *TdSAT1* (**D**), in the roots of two durum wheat cultivars grown under control well-watered (CW), control stressed (CS), mycorrhized watered (MW), and mycorrhized stressed (MS) conditions. Etrusco is represented by white bars and Svevo by black bars. Data are presented as the mean ± SE (*n* = 5) and significant differences between the Etrusco and Svevo plants are indicated as follows: * *p* < 0.05. Different letters indicate significant differences between conditions within a genotype (uppercase letters for Etrusco and lowercase letters for Svevo) (Tukey’s test, *p* < 0.05). An absence of letters means that there were no significant differences between conditions. The two-way ANOVA results are reported in Appendix A.

## Data Availability

The raw data supporting the conclusions of this article will be made available by the authors, without undue reservation, to any qualified researcher.

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
