# Peer review of "Arbuscular Mycorrhizal Symbiosis Differentially Affects the Nutritional Status of Two Durum Wheat Genotypes under Drought Conditions"

_plants, 2022, doi:10.3390/plants11060804_

Round 1

Reviewer 1 Report

The paper deals to an evaluation of the effect of mycorrhization on two durum wheat genotypes, in relation to two different water levels. The manuscript is generally well written, with some minor spells to be improved. The experiments was carried out under controlled conditions. Durum wheat cultivar Svevo has been previously investigated under comparable conditions for different quality traits. Further, the same genotype (Svevo) genome was recently sequenced. However, some doubts are following indicated.

The combination of the four treatments reported in Figures (1-5) can be improved. In particular, the hystograms should refer to the two genotypes under the two water regimes, with the two AMF treatments in couple (white and black). This would help the reader to appreciate the effect of the interaction of the AMF x Water for the two genotypes, that is the aim of the paper, as the authors proposed. This refer to a style effect; the authors are not obliged to receipt this, but an improved should be carried out.

As for PCA, the two components are not well described and an hypothesis on these is not clear. Each component should be label referring to an effect or an hypothetical source of variation and then properly discussed. The authors mostly refer to multivariate analysis and this is not clear. Could they provide the multiple regression analysis that generate the PCA, also in supplementary? This is a critical point.

Finally, it is interesting to observe the outcomes on S-uptake; this can lead to several effects, such as on quality. Did the authors consider this (e.g. on protein content and composition)? An integration on the scientific background is also recommended both at introduction and in discussion.

Minor questions:

Line 546: is the AMF product a commercial one or was this prepared on demand?

Author Response

Reviewers' Comments to the Authors:

Reviewer 1

The paper deals to an evaluation of the effect of mycorrhization on two durum wheat genotypes, in relation to two different water levels. The manuscript is generally well written, with some minor spells to be improved. The experiments was carried out under controlled conditions. Durum wheat cultivar Svevo has been previously investigated under comparable conditions for different quality traits. Further, the same genotype (Svevo) genome was recently sequenced. However, some doubts are following indicated.

Author's response: We thank the reviewer for the comment.

The combination of the four treatments reported in Figures (1-5) can be improved. In particular, the hystograms should refer to the two genotypes under the two water regimes, with the two AMF treatments in couple (white and black). This would help the reader to appreciate the effect of the interaction of the AMF x Water for the two genotypes, that is the aim of the paper, as the authors proposed. This refer to a style effect; the authors are not obliged to receipt this, but an improved should be carried out.

Author's response:  We thank the reviewer for the comment. We reported the details about the interaction between variables in the supplemental files, where the reader can specifically check the two-way ANOVA output. We chose to present data reported in the MS because we aimed to monitor the differences between genotypes under the different treatments. Since two-way ANOVA output revealed that variables interact significantly for some tested parameters and not for others, we decided to present all the data highlighting the comparison between genotypes. All the information concerning multiple comparison analysis has been reported in the text. To explain better this point, we added the following paragraph in the M&M section (4.5):

“Due to the multidimensional nature of the dataset, the trait variations between the two cultivars were analyzed by principal component analysis (PCA) to identify potential morphological and physiological traits associated or not with AM symbiosis under drought stress. Experimental data were subjected to two-way analyses of variance (ANOVAs). Firstly, Two-way ANOVAs were performed within genotypes to evaluate the significance of water status, the AMF treatment, and the interaction of water status × AMF treatment. Secondly, Two-way ANOVAs were performed within water conditions to evaluate the significance of genotype, the AMF treatment, and the interaction of genotype × AMF treatment. Considering that the interaction among variables differed among the parameters analyzed we aimed to highlight the difference between the two cultivars by providing a pair comparison analysis under each treatment in the figures. Principal component analysis and two-way analysis of variance (ANOVA) were conducted using Past3 software.”

As for PCA, the two components are not well described and an hypothesis on these is not clear. Each component should be label referring to an effect or an hypothetical source of variation and then properly discussed. The authors mostly refer to multivariate analysis and this is not clear. Could they provide the multiple regression analysis that generate the PCA, also in supplementary? This is a critical point.

Author's response:  We thank the reviewer for the comment. The multivariate analysis could be carried out by different models including a factorial analysis (such as PCA) and by regression analysis (such as multiple regression analysis). It would have been interesting to make the multiple regression analysis. However, in our study, we selected a PCA model and a not multiple regression analysis. Furthermore, the software we used did not provide information about multiple regression analysis. We modified the PCA output in the MS to enhance the clarity of the results.

Finally, it is interesting to observe the outcomes on S-uptake; this can lead to several effects, such as on quality. Did the authors consider this (e.g. on protein content and composition)? An integration on the scientific background is also recommended both at introduction and in discussion.

Author's response: We thank the reviewer for the comments. As suggested by the reviewer, we have added the following content in the introduction, line 84-90:

“It is well known that S nutrition may affect dough and baking quality through its effect on wheat grain protein composition [50]. Indeed, most of the sulfate taken up by the plant is used for the synthesis of sulfur-containing amino acids, cysteine (Cys) and methionine (Met) [51], which are the major component of proteins [52], which in turn represent about 20% of the grain and play a key role in the dough functionality. Particularly Cys, being responsible for the disulfide bond in the proteins, helps to improve the elasticity of gluten, which affects the processing quality of wheat [53, 54].”

and we added the following paragraph in the discussion, line 510-511:

“and in wheat also play an essential role in the dough functionality, thus affecting bread and pasta making quality [54]”

Minor questions:

Line 546: is the AMF product a commercial one or was this prepared on demand?

Author's response: It is a monospecific inoculum composed by Funneliformis mosseae (formerly Glomus mosseae, BEG 12) propagules. This inoculum and the relative carrier used in non-mycorrhizal condition are both commercial products purchased from MycAgrolab France.

Reviewer 2 Report

I wish to thank the author for his excellent manuscript. The topic of the manuscript is scientifically interesting. The obtained results are very good and will help a lot in better understanding of certain processes that take place in plants in dry conditions. 

Author Response

We appreciate the positive feedback from the reviewer

Reviewer 3 Report

plants-1586845 Review Report

This paper tried to clarify the effect of AMF symbiosis on drought tolerance of durum wheat. Although I admit the topics is opportune and significant, the current manuscript contains many difficulties in experimental design and in analytical methods to be accepted.

How many pots were prepared for each treatment? It is described that 5 plants from each treatment were sampled. If those 5 plants were grown in a same large pot, they cannot be treated as replicated samples (pseudo replication).

The explanatory valuables of PCA analysis should be independent each other to avoid the problem of multicollinearity, however the current PCAs contain variables that seems to be highly correlated (ex. RFW and RL, and SFW and SL for Figure 1).

As for figure 1, I do not understand why authors compared each physiological parameters of two cultivars. Difference responses against the treatments should be compared between the cultivars, not specific parameters such as root length under non-stressed condition. Further, it is described that t-test was applied to evaluate the significance of differences (L628) but multiple comparison should be applied.

Because of the problems mentioned above, the results claimed in this paper are scientifically unreliable. The authors should review their analytical methods and carefully recheck whether the same conclusions can be reached when appropriate methods were applied.

Minor points:

L54 and L56: Avoid using different abbreviations to same phrase.

L91: What were criteria to select these two cultivars?

L554-559: pF value should be indicated to show how severe the drought treatment.

L576: Provider’s name should be accompanied by its location.

L582-L585: Space is necessary between number and unit.

Author Response

Reviewer 3 

This paper tried to clarify the effect of AMF symbiosis on drought tolerance of durum wheat. Although I admit the topics is opportune and significant, the current manuscript contains many difficulties in experimental design and in analytical methods to be accepted.

How many pots were prepared for each treatment? It is described that 5 plants from each treatment were sampled. If those 5 plants were grown in a same large pot, they cannot be treated as replicated samples (pseudo replication). 

Author's response:  The experiment was performed considering 5 independent biological replicates (1 plant/pot). (line 575)

The explanatory valuables of PCA analysis should be independent each other to avoid the problem of multicollinearity, however the current PCAs contain variables that seems to be highly correlated (ex. RFW and RL, and SFW and SL for Figure 1).

Author's response:  We thank the reviewer for the comment. PCA aimed to address how multiple measurements made on each experimental unit affect the total variability observed and the relations among these measurements. Considering that plant growth and nutritional status of plants are defined by several parameters, we aimed to identify the contribution of each parameter (morphometric traits and nutrients) on the total variability of plant growth and nutritional status respectively. The presence or not of collinearity is important to know how parameters behave in response to the variable considered. We modified the PCA outputs in the MS to enhance the clarity of the results.

As for figure 1, I do not understand why authors compared each physiological parameters of two cultivars. Difference responses against the treatments should be compared between the cultivars, not specific parameters such as root length under non-stressed condition. Further, it is described that t-test was applied to evaluate the significance of differences (L628) but multiple comparison should be applied.

Author's response:  We thank the reviewer for the comment. It is not clear what the Reviewer is referring to. We, actually, did multiple comparison analysis (See text and Suppl material). Since two-way ANOVA output revealed that variables interact significantly for some tested parameters and not for others, we decided to present all the data highlighting the comparison between genotypes, under the different treatments, which was the aim of the work. To explain better this point, we added the following paragraph in the M&M section (4.5):

“Due to the multidimensional nature of the dataset, the trait variations between the two cultivars were analyzed by principal component analysis (PCA) to identify potential morphological and physiological traits associated or not with AM symbiosis under drought stress. Experimental data were subjected to two-way analyses of variance (ANOVAs). Firstly, Two-way ANOVAs were performed within genotypes to evaluate the significance of water status, the AMF treatment, and the interaction of water status × AMF treatment. Secondly, Two-way ANOVAs were performed within water conditions to evaluate the significance of genotype, the AMF treatment, and the interaction of genotype × AMF treatment. Considering that the interaction among variables differed among the parameters analyzed we aimed to highlight the difference between the two cultivars by providing a pair comparison analysis under each treatment in the figures. Principal component analysis and two-way analysis of variance (ANOVA) were conducted using Past3 software.”

Because of the problems mentioned above, the results claimed in this paper are scientifically unreliable. The authors should review their analytical methods and carefully recheck whether the same conclusions can be reached when appropriate methods were applied. 

Author's response:  We thank the reviewer for the comment. We added sentences clarifying all the Reviewer’s concerns (see comment above).

Minor points:

L54 and L56: Avoid using different abbreviations to same phrase.

Author's response:  We thank the reviewer for pointing this out. The text has been revised (line 54).

L91: What were criteria to select these two cultivars?

Author's response: We thank the reviewer. we added the following paragraph in the introduction (line 96-99), specifying the criteria employed:

“with different genetic backgrounds have been investigated. Therefore, we select a commercial durum wheat variety (Svevo) and an ancient variety (Etrusco, Triticum turanicum turgidum) cultivated in Italy since the time of the Etruscans, from which the name derives.”

L554-559: pF value should be indicated to show how severe the drought treatment.

Author's response:  We thank the reviewer for the comment. To monitor the drought treatment, we selected both soil (substrate water capacity (SWC%) as described in Lehnert et al. [104]) and plant parameters (RWC). Although we are aware that other parameters are important to be considered, we think that the determination of such SWC and RWC are a good indicator of the induced and perceived degree of drought treatment in our experiment.

L576: Provider’s name should be accompanied by its location.

Author's response:  We thank the reviewer for the suggestion. The text has been revised (line 588).

L582-L585: Space is necessary between number and unit.

Author's response:  As suggested by the reviewer, the text has been revised.

Reviewer 4 Report

This manuscript is about a study of the effect of AMF on wheat plants (2 cultivars) grown under drought stress, in terms of plant growth-related traits as well as plant nutritional status. The results of the study indicated that AM symbiosis differentially affected plant growth-related traits under drought in the two cultivars. Moreover, it was evident that drought differentially affected AMF colonization as well as leaf nutritional status under mycorrhizal and control conditions in the two cultivars. This work is very interesting and significant for the readers. Nevertheless, there are a few shortcomings that need to be addressed:

Title of the manuscript: I think that something is missing there, and especially after the word "affects". Maybe the authors wanted to say something like "Arbuscular mycorrhizal symbiosis differentially affects the plant growth and the nutritional status of two durum wheat genotypes upon drought."?

Materials and Methods, paragraph 4.2: Total RNA was extracted from the whole root system, or from specific roots? Please define which roots were selected as samples and why. 

Materials and Methods, paragraph 4.4: Please define which leaves were selected as samples and why. 

 Supplementary Table S7, primers used for Quantitative RT-PCR: Primer pairs designed for TdPT11 and GADPH do not give products on target templates  in Triticum durum. Instead, they do give the desire product in Triticum aestivum for the first gene and in Triticum aestivum and Triticum dicoccoides for the second one. Please check again the respective primer pairs. 

Author Response

Answer to Reviewer 4

This manuscript is about a study of the effect of AMF on wheat plants (2 cultivars) grown under drought stress, in terms of plant growth-related traits as well as plant nutritional status. The results of the study indicated that AM symbiosis differentially affected plant growth-related traits under drought in the two cultivars. Moreover, it was evident that drought differentially affected AMF colonization as well as leaf nutritional status under mycorrhizal and control conditions in the two cultivars. This work is very interesting and significant for the readers. Nevertheless, there are a few shortcomings that need to be addressed:

Author's response:  We thank the reviewer for the positive evaluation of our work.

Title of the manuscript: I think that something is missing there, and especially after the word "affects". Maybe the authors wanted to say something like "Arbuscular mycorrhizal symbiosis differentially affects the plant growth and the nutritional status of two durum wheat genotypes upon drought."?

Authors response: We thank the reviewer for pointing this out and we apologize for the inaccuracy. The title has been modified as suggested.

Materials and Methods, paragraph 4.2: Total RNA was extracted from the whole root system, or from specific roots? Please define which roots were selected as samples and why. 

Authors response:  We thank the review. Total RNA was extracted from the whole root system since in wheat all roots (primary and lateral) can establish AM symbiosis. The collection of all root systems for AM symbiosis evaluation was also done.

Materials and Methods, paragraph 4.4: Please define which leaves were selected as samples and why. 

Authors response: We thank the review. We modify the text accordingly in the M&M section (line 624-626):

“Determination of cation and anion contents was carried out on the 2nd and 3rd leaves as represented of physiological active leave of plants at harvesting time (the flag leaf was harvested for RWC determination)”

Supplementary Table S7, primers used for Quantitative RT-PCR: Primer pairs designed for TdPT11 and GADPH do not give products on target templates in Triticum durum. Instead, they do give the desire product in Triticum aestivum for the first gene and in Triticum aestivum and Triticum dicoccoides for the second one. Please check again the respective primer pairs. 

We checked the primer sequences as suggested by the reviewer and we can confirm they are both correct. We selected primers for TdPT11 from the literature (Fiorilli et al., 2018). For the GAPDH, by performing a blast analysis, the primers used in our work matched with T.aestivum and dicooides, indeed. However, Liu et al., 2021 (https://doi.org/10.1038/s41598-021-83074-7) used GAPDH as references genes in T. durum samples, as well. The primers selected by Liu et al., 2021 matched only with T.aestivum and dicooides sequences, as occur with the primers used in our work. However, before performing the qRT-PCR analysis we checked their products on genomic DNA and cDNA (gel Figure below) from Svevo and Etrusco varieties and for both genotypes we obtained the desired products. We also checked the efficiency of the primers and the melting curves (charts below) by qRT-PCR. As shown in Fiorilli et al., 2018 the TdPT11 was highly induced in AM roots compared to non-mycorrhizal roots. Therefore, we are confident that our results are reliable.

Round 2

Reviewer 3 Report

I am afraid that authors do not understand my point on statistics and failed to properly revise the manuscript.

As for my comments on explanatory variables of PCA, it is not about the purpose of the analysis but simple technical issue. Authors should check VIF values of explanatory variables in PCAs, and in case the value were not low enough they have to omit some of the explanatory values otherwise the results contain some redundancy.

As for the multiple comparison, authors replied they did perform multiple comparison without indicating the method applied. I am afraid they do not understand what multiple comparison is.

In the revised manuscript, it is described that they repeated ANOVA two times replacing main- and sub- effect which is also curious to me.

Accurate application of appropriate statistical methods is inevitable to obtain reliable results.

Author Response

Authors: We thank the Reviewer for further clarification, that was not fully clear for us, indeed. Following his/her suggestion, we determined the VIF values (reported in attachment) for the different dataset matrices.

Considering that variables are highly correlated in matrices 1 and 4, we deleted the PCA analysis from the manuscripts concerning morphological parameters and gene expression. However, variables from matrices 2 and 3 displayed no correlation or moderate correlation; therefore, we decided to maintain the PCA results in the manuscript. Furthermore, several authors often use PCA analysis on nutrient contents in plants under different growing conditions.

To enhance clarification (likely we were not clear enough), we also organized better results and the statistical methods, following the Editor's suggestion in the revised version.
